# A Novel Electrokinetic-Based Technique for the Isolation of Circulating Tumor Cells

**DOI:** 10.3390/mi14112062

**Published:** 2023-11-05

**Authors:** Mohammad K. D. Manshadi, Mahsa Saadat, Mehdi Mohammadi, Amir Sanati Nezhad

**Affiliations:** 1Mechanical Engineering Department, Southern Methodist University, Dallas, TX 75206, USA; 2Department of Biomedical Engineering, Florida International University, Miami, FL 33199, USA; msaad021@fiu.edu; 3Department of Biological Sciences, University of Calgary, Calgary, AB T2N 1N4, Canada; mehdi.mohammadiashan@ucalgary.ca; 4Department of Biomedical Engineering, Schulich School of Engineering, University of Calgary, Calgary, AB T2N 1N4, Canada; amir.sanatinezhad@ucalgary.ca; 5BioMEMS and Bioinspired Microfluidic Laboratory, Department of Biomedical Engineering, Schulich School of Engineering, University of Calgary, Calgary, AB T2N 1N4, Canada

**Keywords:** electrokinetic, induced-charge, electroosmotic flow, circulating tumor cells (CTC), computational fluid dynamics (CFD)

## Abstract

The separation of rare cells from complex biofluids has attracted attention in biological research and clinical applications, especially for cancer detection and treatment. In particular, various technologies and methods have been developed for the isolation of circulating tumor cells (CTCs) in the blood. Among them, the induced-charge electrokinetic (ICEK) flow method has shown its high efficacy for cell manipulation where micro-vortices (MVs), generated as a result of induced charges on a polarizable surface, can effectively manipulate particles and cells in complex fluids. While the majority of MVs have been induced by AC electric fields, these vortices have also been observed under a DC electric field generated around a polarizable hurdle. In the present numerical work, the capability of MVs for the manipulation of CTCs and their entrapment in the DC electric field is investigated. First, the numerical results are verified against the available data in the literature. Then, various hurdle geometries are employed to find the most effective geometry for MV-based particle entrapment. The effects of electric field strength (EFS), wall zeta potential magnitude, and the particles’ diameter on the trapping efficacy are further investigated. The results demonstrated that the MVs generated around only the rectangular hurdle are capable of trapping particles as large as the size of CTCs. An EFS of about 75 V/cm was shown to be effective for the entrapment of above 90% of CTCs in the MVs. In addition, an EFS of 85 V/cm demonstrated a capability for isolating particles larger than 8 µm from a suspension of particles/cells 1–25 µm in diameter, useful for the enrichment of cancer cells and potentially for the real-time and non-invasive monitoring of drug effectiveness on circulating cancer cells in blood circulation.

## 1. Introduction

Particle separation/sorting methods using microfluidics devices have attracted considerable attention due to their applications in label-free and contact-free disease diagnosis [1,2,3,4]. Several microfluidic devices have been developed to establish different strategies for the manipulation and isolation of bioparticles based on inertial [5,6], optical [4] magnetophoretic [7], dielectrophoresis (DEP) [8,9,10], thermophoresis [11], and acoustophoretic forces [12]. For example, particle manipulation in inertial microfluidics is based on the balance between inertial lift forces and dean forces [13,14], while magnetophoretic particle manipulation is based on controlling the movement of particles by applying an external magnetic field [15].

However, most of these microfluidic devices are very complex [15]. An efficient device for the manipulation of bioparticles needs to be simple in function and cost-effective while offering a high integration capability to make it an operational biomedical tool [16].

The induced-charge electrokinetic (ICEK) method has been emerged as a promising method for sorting of particles and cells in complex biofluids [17,18,19]. When an electric field is applied around a polarizable surface immersed in an ionic fluid, it results in an induced diffuse charge on the surfaces. This charge distribution on the surface leads to the formation of a non-uniform electric double layer (EDL) where negative ions from the electrolyte are attracted to the positive side and positive ions are attracted to the negative side. The consequence of forming a non-uniform EDL is the creation of micro-vortices (MVs) around the surface [20,21].

The ICEK method based on the induction of MVs has been used to produce different applications within microfluidics devices [22]. Different ICEK micromixers were introduced within microfluidic devices using a large number of triangular-shaped polarizable surfaces [23] or a pair of conducting triangle hurdles [22,24]. The ICEK method has also been utilized for the manipulation of conducting particles inside a fluid chamber [25], and fluid pumping, as a result of the motion of conducting particles (Janus microparticles in NaCl solutions) in response to the charge induced by the electric field [26]. Parameters affecting the transport of microparticles under the effect of an electric field was consequently studied in greater detail to analyze the motion of Janus and polarizable particles in AC and DC electric fields [20,27,28]. ICEK-based particle manipulation has been specifically used for the trapping and concentrating of different particles in specific positions within microchannels [16,17,29,30,31,32]. Particles can be trapped in induced MVs produced by ICEKs in a fluid flow generated by placing conducting strips on one or two walls of a microchannel [33,34].

Herein, the capability of ICEK MVs to separate and enrich circulating tumor cells (CTCs), with their heterogeneity in size, is demonstrated. CTCs are cancer cells released from primary tumors, which flow through the vascular system and metastasize to target tissues, spreading the cancer in distant organs [35]. The CTCs’ diagnostic potential stems from the MVs capacity to non-invasively detect cancer, track treatment effectiveness, offer valuable prognostic insights, and contribute to cancer research and drug development endeavors [36].

Isolation of CTCs is very challenging due to their very low numbers in the blood among many other red blood cells (RBCs) and white blood cells (WBCs) [6]. For the first time, we demonstrate their capability to trap cancer cells around conducting hurdles within a fluid flow under the effect of a simple DC electric field. Following the validation of our numerical model, different polarizable hurdle prototypes are examined to determine the effect of different parameters including electric field strength (EFS), cell diameter, and the microchannel wall’s zeta potential, on the trapping efficacy. This investigation provides new insight into CTC separation within microfluidics with exceptional functional characteristics such as simple design, easy and low-cost fabrication, rapid, minimal power requirement, high integration capability, and portability, compared to other microfluidic-based CTC separation kits. The capability for the real-time entrapment and release of CTCs can also open new avenues for the clinical application of this technique for the rapid enumeration of CTCs in the patient’s blood and characterizing the efficacy of different cancer therapeutics.

## 2. Governing Equations

The interaction between the electric field and microchannel walls induces an electroosmotic flow (EOF) in the microchannel. The negatively-charged surface attracts the electrolyte counter ions and creates an electric double layer (EDL) near the microchannel walls [22]. To determine the flow field in the microchannel, momentum (Navier–Stokes) and continuity equations (Equations (1) and (2)) are solved for the incompressible fluid flow in the presence of an external electric field [22,37].
(1)$$\rho [\frac{\partial u}{\partial t}+u\cdot \nabla u]=-\nabla P+\mu \nabla^{2}u+\rho_{e}E$$
(2)∇⋅u=0
where equation *ρ* (kg/m^3^) denotes fluid density, **u** (m/s) represents flow velocity, P (Pa) is pressure, $\rho_{e}$ (C/m^3^) is net electric charge density, **E** (V/m) denotes local applied electric field calculated from $E=-\nabla \phi_{e}$, where $\phi_{e}$ (V) is the applied electric potential, and *ρ_e_***E** is the Coulomb force. This force is negligible when the EDL is a thin layer and the local net charge density is present only in the EDL. In this condition, Helmholtz–Smoluchowski wall slip velocity (**u**_slip_) boundary condition can be applied to the microchannel wall (Equation (3)) [22,38,39]:(3)$$u_{\mathrm{slip}}=\mu_{\mathrm{eo}}E_{t}$$
where *µ*_eo_ (m^2^/(V·s)) denotes the electroosmotic mobility and **E_t_** (V/m) represents the tangential electric field strength. The electroosmotic mobility is defined using Equation (4) [39].
(4)$$\mu_{\mathrm{eo}}=-\frac{\varepsilon_{0}\varepsilon \zeta }{\mu }$$
where *ε*_0_ is free space permittivity (F/m), *ε* is electrolyte relative permittivity, and *ζ* (V) represents the wall’s zeta potential. *ζ* is constant for non-conducting walls but for a conducting wall, a non-uniform zeta potential is induced around the conducting object due to the non-uniform charge accumulation on the surface. Wu and Li [22] modeled the induced zeta potential (IZP) distribution on the conducting surfaces (*ζ_i_*) defined in Equation (5) [22].
(5)$$\zeta_{i}=-\phi_{e}+\phi_{c}$$
where *ϕ_e_* is the external electric potential and *ϕ_c_* is a constant correction potential derived from Equation (6) [22].
(6)$$\phi_{c}=\frac{\int_{s}\phi_{e}\mathrm{dA}}{A}$$
where A is conducting surface area. Newton’s second law is then considered, as in Equation (7), to trace a particle in a medium.
(7)$$\frac{d(m_{p}V)}{dt}=\sum F_{t}$$
where **V** is the particle velocity and **F***_t_* is the exerted forces on the particle. This equation shows that the rate of particle’s momentum change (*m_p_***V**) equals the net force acting on particle such as gravity, electrostatic, and hydrodynamic forces. Herein, the gravitational effects are negligible because cell and electrolyte densities are almost the same. In addition, the electrostatic force can be also neglected due to the thin EDL around the cells compared to their diameter [40,41,42]. Drag force is the acting hydrodynamic force on the particles from the flow field and it is obtained from Stokes law, defined as Equation (8) [43].
(8)$$F_{drag}=\frac{18\mu }{\rho_{p}d_{p}^{2}}m_{p}(u-V)$$
where *ρ_p_* (kg/m^3^) is the particle density, *d_p_* (m) is the particle diameter, *m_p_* (kg) is the particle mass, and **u** (m/s) represents the flow velocity.

In the simulations, the electric field was then adjusted to a maintain maximum applied electric field strength below 100 V/cm to avoid a rise in temperature due to the electric current passing through the buffer (Joule heating effect) and prevent damage to living cells [34,44].
(9)$$\rho C_{p}(\frac{\partial T}{\partial t}+(u\cdot \nabla )T)=\nabla \cdot (k_{m}\nabla T)+S$$
(10)S=σ(E.E)
where *C_p_* (J/K) is the heat capacitance, *T* (K) is the temperature, *k_m_* (W/m·K) is the fluid heat conductivity, *σ* (S/m) is the buffer electrical conductivity, and *S* (J/m^3^) is the Joule heating source term. According to the literature, the critical applied electric field that causes cell damage in the microfluidics devices is 600 (V/cm) [18,34,44,45].

## 3. Numerical Method

COMSOL Multiphysics software (a FEM based software, V.5.4) is employed to calculate the equations governing the model. Laminar flow, electrostatics, and particle tracing for fluid flow modules are coupled together to solve the considered model. First, the independency of the results from the grids is verified (e.g., a number of about 62,000 triangular elements confirmed the independency of the simulation results from the mesh size with the error less than 2%). Second, prior to modeling the cell trapping, the reliability of the numerical simulations is validated by comparing the results with theoretical results obtained from the literature. Squires and Banzant [46] proposed the analytical Equation (11) for the IZP distribution over a polarizable surface in a two-dimensional (2D) circular cylinder and in a uniform applied electric field.
(11)$$\zeta_{i}(\theta )=2E_{0}a\cos(\theta )$$
where “*a* (m)” is the cylinder radius, *E*_0_ (V/m) is the applied uniform electric field, and *θ* is shown in Figure 1A. For instance, Equation (11) is used to determine the IZP on a 2D circular conducting cylinder with the radius of *a* = 50 µm within a 40 V/cm electric field inside an enclosed chamber with *L* = *W* = 1 mm (Figure 1A). The IZP on the cylinder surface generates MVs around the surface (Figure 1B). The numerical results of the IZP distribution within a uniform applied electric field are in good agreement with the theoretical results proposed by Squires and Banzant (Figure 1C) [46].

## 4. Results and Discussion

The validated numerical model is used to assess the capability of induced MVs to separate CTCs (15–30 µm) from red and platelet cells (1–8 µm) [47]. A conducting hurdle is placed in an EOF, and the effects of the key parameters, including hurdle geometry, EFS, and wall zeta potential for particle trapping within the induced MVs, are evaluated. The microchannel model was first employed (Figure 2A). Different hurdle shapes, including the microchannel proposed by Wu and Li [24] as well as cylindrical, triangular, and rectangular hurdles, were employed to find the most efficient configuration for particle/cell separation (Figure 2). The boundary conditions (BCs) and material properties for these configurations are listed in Table 1, and their trapping efficiency is evaluated in a 100 V/cm EFS for different sizes of cells. The results show that only the induced MVs around the rectangularly shaped hurdle lead to full cell entrapment due to the higher distance between the induced frontal and rear MVs around the hurdle (Figure 2).

To investigate particle motility in the MVs around the rectangular-shaped hurdle, two particles of 10 µm in diameter are subject to three different electric fields of 50 V/cm, 71 V/cm, and 100 V/cm and particles are traced within the microchannel (Figure 3). The results show that the particle reaching the MVs has three possible routes of FD1, FD2, and FD3. In the FD3 route, the particle enters the rear MVs and escapes (Figure 3B). In the FD2 route, the particle enters near the wall EOF and escapes (Figure 3C). Finally, in the FD1 route, the particle returns to the front of the hurdle near the frontal MVs and is trapped (Figure 3D). Since the drag force is dependent on the size of particles or cells, the drag force applied on the particle needs to be the same both near the wall EOF and rear MVs in order to trap a particle within the frontal MVs and return it to the front of the hurdle. Appendix A contains the corresponding raw data from Figure 3.

The effects of the EFS and particle/cell diameter on the efficacy of particle trapping in the MVs are further investigated. Figure 4A shows the effect of increasing the EFS on the entrapment of cells with different diameters of 5, 10, and 15 µm. The results show that the trapping fraction increases with the rise in the EFS, where above 90% of 10 µm cells are trapped in EFSs higher than 85 V/cm. The main reason for the escaping of some particles is the EOF flow near the wall where these particles enter the near-wall diffuse layer and flee from the MVs. Such partial entrapment was shown to be reduced for smaller cells due to their smaller response to the drag force. Appendix A contains the corresponding raw data of Figure 4A.

Figure 4B shows cell trapping with a diameter of 1 µm to 25 µm under 65, 75, and 85 V/cm electric fields. Above 90% of cells with a diameter larger than 7 µm are trapped with an EFS of 85 V/cm. Moreover, all cells smaller than 4 µm in diameter escape from the MVs. Consequently, cells with a diameter larger than 8 µm can be separated from platelets of 1–3 µm [47]. In addition, decreasing the strength of the electric field increases the chance of the cell escaping for larger cells. For instance, MVs produced by an EFS of 75 V/cm trap the cells larger than 15 µm but allow the cells smaller than 10 µm in diameter to escape, something that enables the separation of CTCs from other cells in the blood. Appendix A contains the corresponding raw data from Figure 4B.

To reduce the effect of near-wall EOF and the resultant particles escaping through this layer, the microchannel wall zeta potential (|*ζ*|) can be manipulated using ionic surfactants or electrolytes [48]. Figure 5 shows that |*ζ*| influences the trapping of 10 µm cells in an 85 V/cm applied electric field. Although decreasing the |*ζ*| value results in a better cell trapping, it increases the trapping time. For example, particles are trapped after *t* = 10 s when |*ζ*| = 10 mV but this time decreases to *t* = 4 s at |*ζ*| = 30 mV. Therefore, decreasing the zeta potential magnitude of the wall results in a higher trapping efficiency. Using electrolytes such as sodium chloride (NaCl) in glass microchannels decreases the zeta potential magnitude of the wall and increases the chance of cell trapping [48]. Appendix A contains the corresponding raw data from Figure 5E.

## 5. Conclusions

CTCs are very rare cancer cells that undergo several metastatic stages prior finding their way into the blood. Several techniques have been developed for the separation and enrichment of CTCs which are useful for the diagnosis of tumor stage and assisting in therapeutic decisions about patients with malignancies. Despite the success of existing platforms for the separation of CTCs, the clinical use of these assays is still laborious. Although marker-free isolation devices are developed to avoid complicated assays, such devices enrich the heterogeneous population of cells and separate RBCs and WBCs as false positives.

To overcome the bottlenecks of the existing technologies for the separation of rare circulating tumor cells in the blood, this work offers a simple microfluidic platform with a fast response for the effective trapping of CTCs with a 100 cell/s throughput. The MVs induced around polarizable hurdles in a microchannel with an EOF are employed to entrap the cells. We used our validated numerical model to determine which configuration of the conducting hurdles provides the most efficient cell trapping performance in the microchannel. The results show that the MVs produced around the rectangular-shaped hurdles provided the highest trapping performance for cells. We further studied the effect of the EFS, cell diameter, and zeta potential of the microchannel wall on the trapping efficiency. Although the increase in the EFS leads to more powerful MVs around the hurdle surface, it also raises the EOF. Therefore, an appropriate EFS should be applied to balance these two effects for effective cell trapping. Applying an EFS greater than 75 V/cm in the microchannel resulted in the entrapment of above 90% of cells with a diameter larger than 15 µm. Such a trapping performance can be highly applicable for separating CTCs (15–30 µm) from blood cells. Increasing the EFS to 85 V/cm also resulted in the trapping of particles larger than 8 µm from a suspension of particles/cells with diameter of 1 µm to 25 µm. Moreover, reducing the zeta potential of the wall decreased the effect of the EOF on the escaping cells. The results show that reducing the zeta potential of the wall (e.g., using electrolytes) increased the possibility of particle trapping. The proposed platform has high biocompatibility with cells due to its lower EFS compared with previous DEP methods. In addition, Joule heating is negligible in this platform because the maximum EFS across the microchannel is below 100 V/cm, which is less than the critical Joule heat (EFS about 600 V/cm). The capability for the real-time entrapment and release of CTCs opens avenues for the rapid enumeration of CTCs in the patient’s blood and for characterizing the efficacy of different cancer therapeutics.

## Figures and Tables

**Figure 1 micromachines-14-02062-f001:**
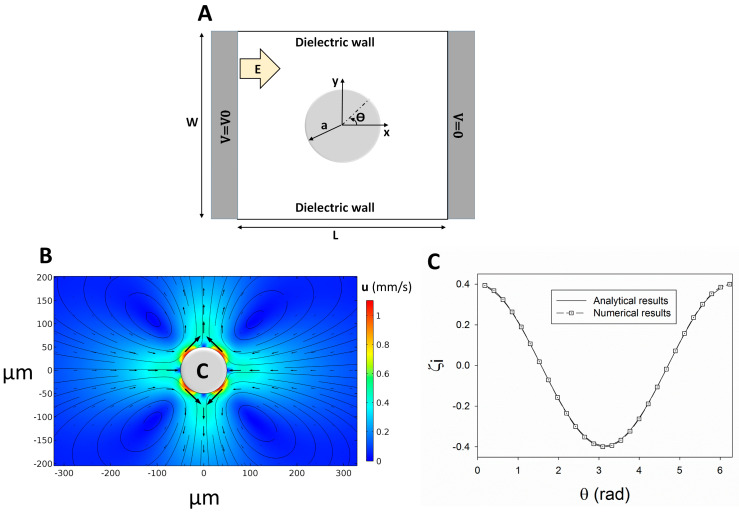
Validation of the numerical simulation for determining the induced zeta potential (IZP) over a polarizable surface in a two-dimensional (2D) circular cylinder under a uniform applied electric field. (**A**) Schematic of the studied cylinder, (**B**) IZP distribution on the conducting cylinder, and (**C**) comparing the numerical data against analytical results from Squires and Banzant [46].

**Figure 2 micromachines-14-02062-f002:**
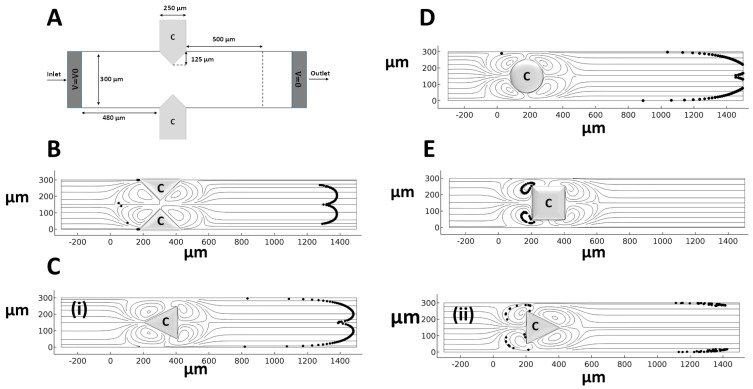
Different shapes and configurations of hurdles within the microchannel for assessing the efficacy of bioparticle trapping (*t* = 5 s after releasing 100 bioparticles). (**A**) The microchannel model of Wu and Li [24]. (**B**) Two triangular conducting hurdles, (**C**) triangular hurdles with different directions of (**i**) and (**ii**), (**D**) a cylindrical hurdle, and (**E**) a rectangular hurdle.

**Figure 3 micromachines-14-02062-f003:**
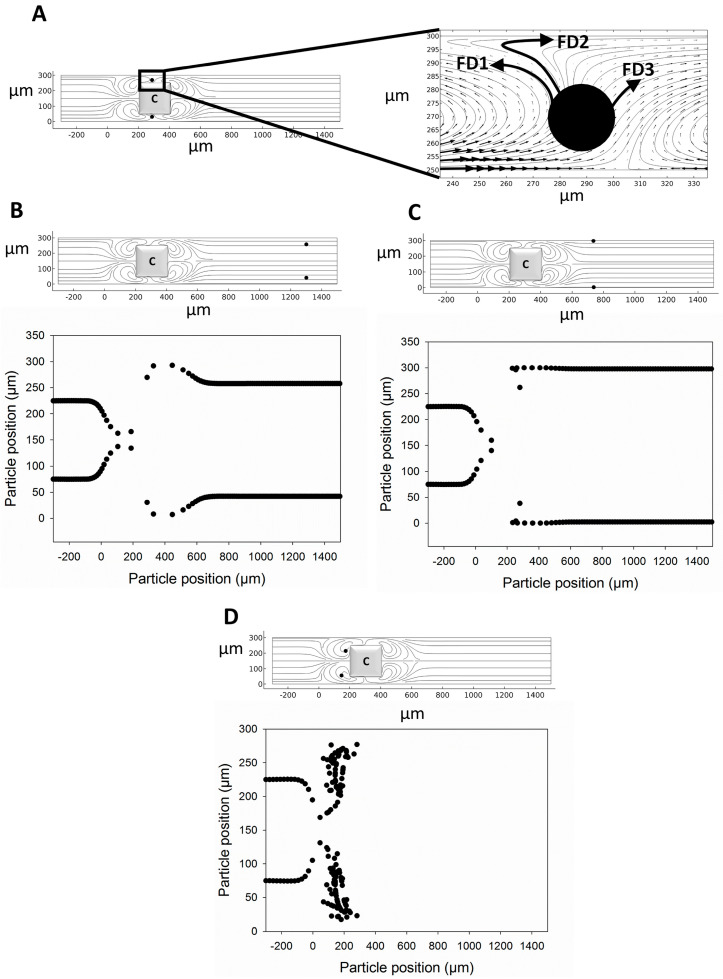
The trajectory of two particles of 10 µm diameter in MVs at different electric field strengths. (**A**) Three route choices for a particle of 10 µm diameter to move through MVs, (**B**) cell escape in 50 V/cm EFS, (**C**) cell escape in 71 V/cm EFS, and (**D**) cell entrapment in 100 V/cm EFS.

**Figure 4 micromachines-14-02062-f004:**
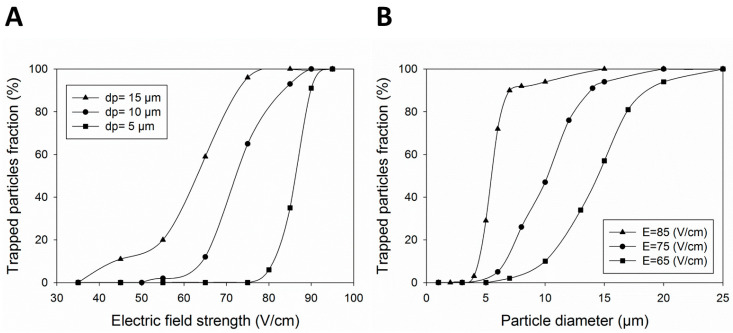
The effect of different (**A**) EFSs and (**B**) cell diameters on trapping efficacy for the rectangular-shaped hurdle.

**Figure 5 micromachines-14-02062-f005:**
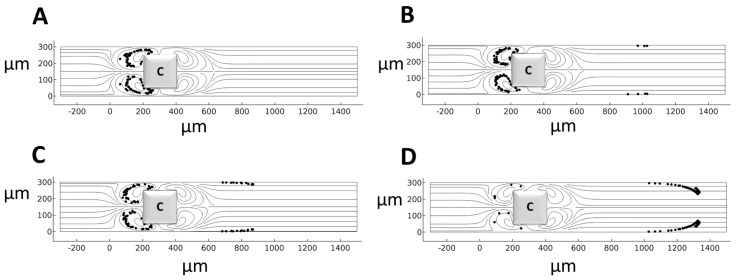
The effect of |ζ| on the trapping of 10 µm cells (*t* = 5 s). (**A**) |*ζ*| = 30 mV, (**B**) |*ζ*| = 50 mV, (**C**) |*ζ*| = 70 mV, (**D**) |*ζ*| = 100 mV, and (**E**) the trapped particle fraction at different |*ζ*| magnitudes.

**Table 1 micromachines-14-02062-t001:** Material properties and boundary conditions (BCs) for different configurations of hurdles within microchannels for cell entrapment under an electric field [24].

Material Properties	
Electrolyte density (*ρ*)	998 kg/m^3^
Viscosity (µ)	0.001 Pa·s
Particle diameter	1–25 µm
Cell density	1050 kg/m^3^
**BCs**	
Non-conducting walls	Slip velocity (Equation (4)) with *ζ* (V)
Conducting walls	Slip velocity (Equation (4)) with *ζ_i_* (V)
Inlet	No viscous stress Electric potential (*V*_0_) Release of 100 particles
Outlet	No viscous stress Zero voltage (ground) Particle escape

## Data Availability

Data is unavailable due to privacy.

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
