# Peer review of "A Novel Electrokinetic-Based Technique for the Isolation of Circulating Tumor Cells"

_micromachines, 2023, doi:10.3390/mi14112062_

Round 1
Reviewer 1 Report
Comments and Suggestions for Authors
The manuscript of Mohammad K.D. Manshadi and Mahsa Saadat (Novel Electrokinetic - based Technique for Isolation of Circulating Tumor Cells) aims to address an important issue of separation of rare cells from complex biofluids, particularly, the selection of circulating tumor cells (CTCs) from the blood. The manuscript is well-written and provides very detailed modeling and simulation of directed micro-vortices of cells/particles around the polarizable hurdles of different shapes in a complex fluid environment using COMSOL software. The impacts of electric field strength, particle diameter, and wall zeta potential magnitude on the trapping efficacy for the rectangular hurdle are the main outcomes of the study. The literature review incorporated in the text provides an excellent introduction to the field. Overall, this is an interesting and well-designed theoretical study, which I recommend for publication in the Micromachines journal with a minor revision. Below, I have provided a couple of suggestions for data presentation and analysis:
1. For better visualization of the simulation, it will be important to provide some numeric raw data in the supplemental material.
2. Does any correlation exist between the number (percentage) of cancer cells/particles in the complex fluid and the trapping efficacy for the rectangular hurdle at the particular value of electric field?
3. What is the potential rate of analysis/sorting (number of cells per sec) in this potential instrument? Please include this in the discussion.
Author Response
We sincerely thank the reviewer for the constructive suggestions and comments, which significantly improved the quality of our manuscript. Based on the recommendations, modifications have been made throughout the text, highlighted in green. Attached here is our reply to the questions asked by the reviewer.

Reviewer 2 Report
Comments and Suggestions for Authors
The authors present numerical work to study the capability of MVs to manipulate CTCs and trap them in the DC electric field. Overall, the work is interesting, and the results are promising. Still, the paper's structure, organicity, and the way the content is argued make it unconvincing from a technical/scientific point of view. For example, the research objectives and future developments need to be clarified. The organization of the paragraphs is fine, but the authors should better elaborate on the introduction and enhance the conclusion. The work is written in clear English, but very basic and not very technical. Good image quality and resolution.
Here, are my comments on the manuscript:
· The introduction lacks relevant information and there is no complete state of the art (see, i.e, Multiplexed Near-Field Optical Trapping Exploiting Anapole States. ACS nano., 2023). For example, the authors briefly introduce Microfluidic devices without explaining what the technique of microfluidics consists of, what are the advantages of these devices over conventional ones, why they are used and successful, and their limitations. On the other hand, good and specific Theoretical references related to the technique based on electrokinetics.
· The introduction is messy and confusing; there is no clear connection between one topic and another.
· No references and identification about fluid flow.
· The electric field was then adjusted to maintain maximum applied electric field strength below 100 V/cm to avoid Joule heating and prevent damage to living cells [36, 41] “. The authors should report the equation and describe it in detail.
· The authors should justify more and more clearly what CTCs are and their diagnostic potential in the introduction section. Limited information about this and the technologies adopted to separate and enrich CTCs exists. Also, it is necessary to emphasize that CTCs provide information not only on the current status of the cancerous disease but also on the possible manifestation of metastasis. I suggest the following paper “Follain, G., Herrmann, D., Harlepp, S., Hyenne, V., Osmani, N., Warren, S. C., ... & Goetz, J. G. (2020). Fluids and their mechanics in tumour transit: shaping metastasis. Nature Reviews Cancer, 20(2), 107-124.”
· “Several microfluidic devices have been developed to establish passive and active strategies for the manipulation and isolation of bioparticles based on inertia forces [6, 7], magnetophoretic [8], dielectrophoresis (DEP) [9-11], and acoustophoretic”. It is limiting to list only these strategies without going into the details of each; we also often resort to using optical, thermocapillary, electromagnetic, and acoustic forces on the surface. See, e.g. “M. McEnery, A. Tan, J.D Alderman, J. Patterson, S.C.O’Mathuna, J.D. Glennon. Liquid chromatography on-chip: progression towards a μ-total analysis systemPresented at the SAC 99 Meeting Dublin, Ireland, July 25–30, 1999. Analyst, 125(1), 25-27, 2000”.
Author Response

(The authors gave the same response as above.)

Round 2
Reviewer 2 Report
Comments and Suggestions for Authors
The Authors have modified the manuscript according to the Reviewers suggestions.